# LapBoost: Graph Laplacian Regularized Gradient Boosting for Semi-Supervised Learning

## Abstract

Semi-supervised learning (SSL) has achieved remarkable success in high-dimensional domains through consistency-based methods, yet effective SSL approaches for structured tabular data remain critically underexplored. While gradient boosted decision trees dominate supervised tabular learning, no systematic framework exists for integrating graph-based regularization with gradient boosting to exploit manifold structure in unlabeled data. We introduce LapBoost, the first principled integration of graph Laplacian regularization with modern gradient boosting frameworks, combining LapTAO (Laplacian-regularized Tree-based Alternating Optimization) as base learners within an XGBoost-style ensemble. Our approach enables systematic exploitation of unlabeled data through manifold assumptions while preserving the sequential error correction of gradient boosting. Through comprehensive evaluation across 180 experimental conditions spanning tabular, text, and high-dimensional datasets, we demonstrate that LapBoost achieves statistically significant improvements over supervised baselines in label-scarce regimes, with particularly strong performance on structured data where manifold assumptions hold. Critically, our analysis reveals fundamental complementarity between SSL paradigms: graph-based methods like LapBoost excel on structured data with prominent manifold structure, while consistency-based methods like FixMatch dominate high-dimensional data with rich augmentation possibilities. This finding provides the first systematic characterization of when different SSL approaches should be applied, offering practical guidance for method selection based on data characteristics.

**Keywords:** semi-supervised learning, gradient boosting, graph Laplacian regularization, manifold learning, tree ensembles, XGBoost, graph-based learning

## 1 Introduction

Semi-supervised learning (SSL) has become an essential approach by using large amounts of unlabeled data alongside a limited number of labeled samples to improve model performance (Chapelle et al., 2006; Zhu & Goldberg, 2009). The fundamental idea of SSL is that unlabeled data contains valuable structural information that can enhance learning when labeled examples are scarce, a common scenario in domains such as medical diagnosis, fraud detection, and natural language processing, where obtaining labeled data is expensive or time consuming.

Most recent SSL research has largely focused on two dominant paradigms: consistency-based methods and graph-based approaches. Consistency-based methods, such as FixMatch (Sohn et al., 2020) and its variants (Zhang et al., 2021; Wang et al., 2023), achieve remarkable success on high-dimensional data by enforcing prediction consistency under data augmentation. For example, Fix-Match achieves a precision of 94.93% on CIFAR-10 with only 250 labeled samples. However, these methods are highly dependent on strong and meaningful data augmentations, which are often not available for structured tabular data (Somepalli et al., 2021).

Graph-based SSL methods, rooted in manifold learning theory (Belkin et al., 2006), promote smooth predictions over a data graph and is well suited to domains with reliable neighborhood structure. Despite this fit and their strong theoretical, graph-based methods have received less attention in recent SSL work, especially in combination with modern ensembles. This gap is particularly noticeable for

tabular data, where tree-based models such as XGBoost (Chen & Guestrin, 2016) and LightGBM (Ke et al., 2017) dominate supervised benchmarks but lack effective SSL extensions.

Gradient boosted decision trees (GBDTs) represent the prevailing approach for tabular machine learning, often outperforming deep models on structured data (Borisov et al., 2022) due to their native handling of heterogeneous features, robustness to missing values, and interpretability. Yet, integrating graph-based regularization with gradient boosting for SSL remains unexplored. Existing SSL adaptations of boosting mostly rely on pseudo-labeling (Natekin & Knoll, 2013) or margin-based methods (Wang, 2011) that do not exploit the sequential residual fitting dynamics of modern GBDTs while systematically using unlabeled data.

This gap is significant given the complementary strengths of graph-based SSL and tree ensembles: graphs capture manifold structures in heterogeneous feature spaces, while tree ensembles refine complex decision boundaries via sequential error correction. However, no prior work has systematically combined these paradigms to leverage their synergistic potential for semi-supervised learning.

We address this gap with LapBoost, an SSL framework that integrates graph Laplacian regularization into gradient boosting via principled base learners. Our three key contributions are as follows:

**(1) Systematic integration of graph Laplacian regularization with gradient boosting**: Building on prior margin based boosting (Wang, 2011; Mallapragada et al., 2009) and LapTAO for single trees (Zharmagambetov & Carreira-Perpiñán, 2022), we develop the first comprehensive approach embedding LapTAO within an XGBoost-style ensemble to incorporate manifold structure into gradient boosting, enabling consistent use of unlabeled data throughout boosting.

**(2) LapBoost algorithm using LapTAO as base learner**: We propose a gradient boost framework, replacing traditional CART with LapTAO trees, preserving sequential residual fit while injecting manifold-aware regularization at each step. The result is a unified ensemble that exploits graph structure without altering the optimization behavior that makes GBDTs effective.

**(3) Comprehensive empirical analysis revealing fundamental complementarity**: Across 180 experiments on five diverse datasets, we show that combining graph-based SSL with gradient boosting significantly outperforms existing semi-supervised boosting and state-of-the-art SSL methods.

Figure 1a illustrates that traditional XGBoost relies solely on labeled data, creating rigid decision boundaries. In contrast, LapBoost, which utilizes LapTAO base learners, incorporates both labeled and unlabeled data to produce smoother boundaries that better align with the data's underlying geometry. By integrating manifold structure via graph Laplacian regularization, LapBoost leverages unlabeled data to create more generalizable decision boundaries.

Our evaluation confirms that this approach yields statistically significant improvements over supervised baselines, particularly in low-label settings and on structured data. This offers a practical guideline for selecting semi-supervised learning methods: graph-based methods like LapBoost are ideal for data with a strong manifold structure, while consistency-based methods are better suited for high-dimensional data that can be augmented.

## 2 RELATED WORK

### 2.1 GRAPH-BASED SEMI-SUPERVISED LEARNING

Graph-based SSL methods rely on the *manifold regularization*, which states that data points lie on a low-dimensional manifold where neighboring points share similar labels (Belkin et al., 2006; Zhu et al., 2003). Belkin et al. (Belkin et al., 2006) formalized this idea through *manifold regularization*, adding the graph Laplacian penalty $\mathbf{f}^\top L \mathbf{f}$, where $L$ is the graph Laplacian and $\mathbf{f}$ the label vector, to the supervised loss. Early algorithms, including label propagation (Zhu & Ghahramani, 2002), label spreading (Zhou et al., 2003), and local global consistency (Zhou et al., 2004), explicitly optimize this smoothness criterion.

Recent research has improved graph construction methods and scalability. Wang and Zhang (Wang & Zhang, 2007) proposed adaptive graph learning that jointly optimizes graph structure and label prediction. Liu et al. (Liu et al., 2010) introduced anchor graphs to reduce computational complexity from $O(n^3)$ to $O(nm)$, where $m \ll n$ is the number of anchor points. Graph neural networks

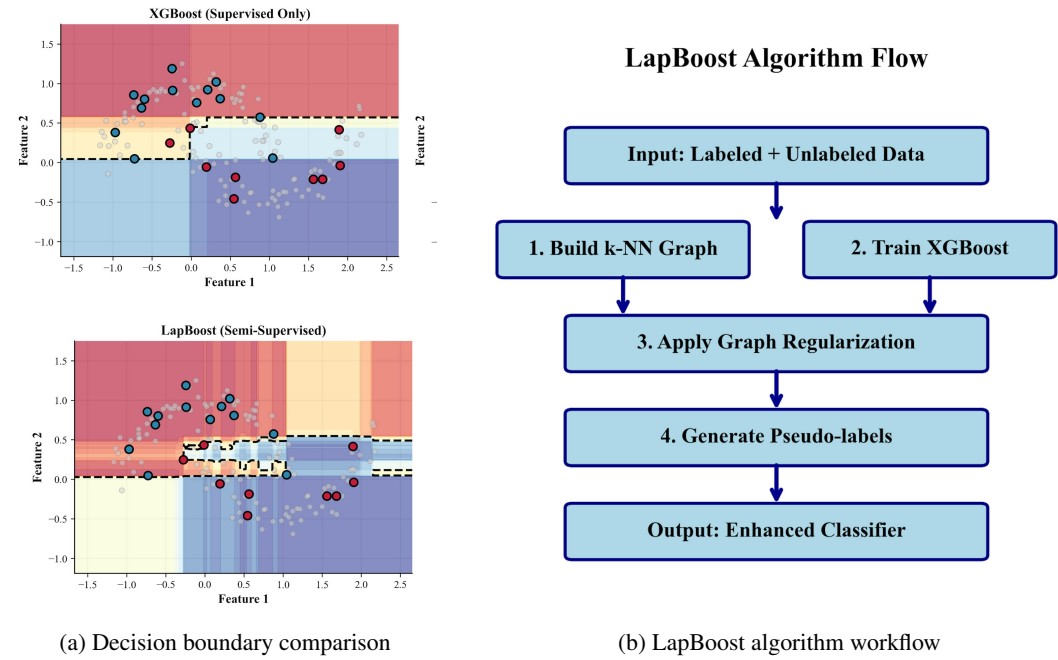

(a) Decision boundary comparison       (b) LapBoost algorithm workflow

Figure 1: LapBoost: Manifold-Aware Decision Boundaries and Framework.

(GNNs) extend these ideas to semi-supervised node classification, with GCN (Kipf & Welling, 2017) and GraphSAGE (Hamilton et al., 2017) demonstrating strong performance on fixed graphs.

However, most graph-based SSL research has focused on homogeneous data or specific domains such as social networks. There is limited work on integrating graph regularization with ensemble methods, particularly for heterogeneous tabular data. Levatić et al. (Levatić et al., 2017) proposed semi-supervised classification trees using graph-based split criteria, but their approach is restricted to single-tree models rather than ensemble techniques.

## 2.2 SEMI-SUPERVISED ENSEMBLE LEARNING

Semi-supervised ensembles rely primarily on self-training or co-training. Mallapragada et al. (Mallapragada et al., 2009) introduced SemiBoost combines boosting with manifold assumptions, but focuses on general boosting frameworks rather than modern gradient boosting implementations. Wang et al. (Wang, 2011) proposed Laplacian Margin Distribution Boosting (LapMDBoost), which attaches a Laplacian constraint to margin-distribution boosting via column generation rather than the residual-based approach characteristic of gradient boosting.

More recent work adopts heuristic pseudo-labeling. Lu et al. (Lu et al., 2021) explored confidence-based pseudo-labeling for random forests, while Jagat et al. (Jagat et al., 2023) applied self-training to XGBoost using threshold-based heuristics. However, these methods lack a principled integration of manifold assumptions and graph regularization.

LapTAO (Zharmagambetov & Carreira-Perpiñán, 2022) is the closest predecessor, which integrates graph regularization into a single decision tree via alternating optimization between label smoothing (solving a linear system) and tree induction. Because it optimizes only one tree, it cannot leverage the residual-fitting dynamics central to gradient boosting, which require sequential residual fitting and additive model construction.

## 2.3 CONSISTENCY-BASED SEMI-SUPERVISED LEARNING

Consistency-based SSL methods enforce invariance of model predictions under data augmentation and have demonstrated outstanding results on high-dimensional datasets. The FixMatch framework (Sohn et al., 2020; Zhang et al., 2021; Wang et al., 2023) exemplifies this approach by generating

pseudo-labels from weakly augmented inputs and applying consistency regularization on strongly augmented counterparts. FlexMatch introduces class-aware thresholds (Zhang et al., 2021), while FreeMatch adapts thresholds over training steps (Wang et al., 2023). DASH (Xu et al., 2021) and SoftMatch (Chen et al., 2023) refine pseudo-label weighting.

A key limitation for tabular data is the lack of semantics-preserving augmentations. Unlike images where rotations and crops preserve semantic content, tabular features often have precise meanings where perturbations alter semantic interpretation (Somepalli et al., 2021). This limitation motivates graph-based approaches that exploit neighborhood structure without relying on augmentations.

## 2.4 Semi-Supervised Learning for Tabular Data

Compared to vision and natural language processing, SSL research on tabular data is underexplored. Early self-training and co-training methods (Nigam & Ghani, 2000) struggle with heterogeneous features and class imbalance common in tabular datasets.

Deep tabular models with SSL extensions, such as VIME, which uses feature masking for self-supervised pretraining (Yoon et al., 2020) and TabNet (Arik & Pfister, 2021), these approaches often underperform compared to tree-based ensembles on small to medium tabular datasets (Borisov et al., 2022; McElfresh et al., 2023).

The recent introduction of TabPFN (Hollmann et al., 2023), a transformer-based model, shows promise for tabular learning but requires substantial pretraining and is limited to specific dataset sizes and types.

Our literature review identifies a clear gap: no previous work systematically integrates graph Laplacian regularization with modern gradient boosting frameworks for SSL. Although LapMD-Boost (Wang, 2011) combines graph regularization with margin-based boosting and LapTAO (Zharmagambetov & Carreira-Perpiñán, 2022) applies graph regularization to single trees, neither addresses the sequential residual fitting and additive model construction intrinsic to gradient boosting.

# 3 Methodology

## 3.1 Problem Formulation and Theoretical Framework

This study addresses the semi-supervised learning problem within the manifold regularization framework (Belkin et al., 2006). Given a labeled dataset $\mathcal{D}_l = \{(x_i, y_i)\}_{i=1}^l$ and an unlabeled dataset $\mathcal{D}_u = \{x_j\}_{j=l+1}^n$ where $x_i \in \mathbb{R}^d$ and $y_i \in \mathcal{Y}$, we aim to learn a predictor $f : \mathbb{R}^d \to \mathcal{Y}$ that leverages the geometry of both labeled and unlabeled data.

Our approach is built upon two fundamental assumptions: (1) the *manifold assumption*, which posits that data lies on or near a low-dimensional manifold embedded in the ambient space, and (2) the *smoothness assumption*, which requires that the target function varies smoothly along the data manifold. These assumptions imply that nearby points should have similar predictions, a notion formalized via graph-based regularization.

LapBoost's core innovation is the systematic integration of manifold regularization into gradient boosting framework, using LapTAO trees as manifold-aware base learners that optimize gradient objectives while respecting geometric structure.

## 3.2 LapBoost Framework Overview

LapBoost extends the traditional gradient boosting paradigm by replacing standard CART (Classification and Regression Tree) trees with base learners that are explicitly regularized by manifold properties. The framework consists of three core innovations: (1) Manifold-Regularized Gradient Boosting Objective, which extends the standard XGBoost objective function to include graph Laplacian regularization terms that enforce smoothness across the data manifold. (2) Gradient-Based Lap-TAO Algorithm, which adapts the alternating optimization procedure of LapTAO to operate directly with gradient and Hessian targets rather than raw labels, enabling direct integration into the boosting framework. (3) Progressive Pseudo-Labeling with Confidence Weighting, this stage involves itera-

tively expanding the training set by incorporating high-confidence pseudo-labels that are seamlessly integrated into the gradient boosting process via sample weighting.

LapBoost aims for each ensemble tree to minimize both the standard boosting loss (fitting gradients) and the manifold regularization loss (respecting geometry), yielding a unified SSL ensemble approach.

Figure 1b illustrates LapBoost's synergistic workflow, and demonstrates how these components work synergistically through four integrated stages. Initially, a k-Nearest Neighbors (k-NN) graph is constructed to capture manifold structure from both labeled and unlabeled data, establishing the geometric foundation for regularization. Subsequently, within each boosting iteration, graph regularization is applied through the gradient-based LapTAO algorithm, ensuring that individual trees not only fit the residuals but also respect the underlying data geometry. Finally, high-confidence pseudo-labels are generated from the ensemble's predictions, progressively expanding the effective training set while maintaining the quality and reliability of predictions through a confidence-weighting mechanism.

## 3.3 GRAPH CONSTRUCTION AND MANIFOLD STRUCTURE

Given the combined dataset $\mathcal{X} = \mathcal{D}_l \cup \mathcal{D}_u$, we construct a weighted undirected graph $G = (V, E, W)$ using adaptive $k$-nearest neighbor with Gaussian similarity kernels and derive the normalized graph Laplacian:

$$w_{ij} = \begin{cases} \exp\left(-\frac{\|x_i - x_j\|^2}{2\sigma^2}\right) & \text{if } j \in \text{kNN}_k(i) \text{ or } i \in \text{kNN}_k(j) \\ 0 & \text{otherwise} \end{cases}$$

$$L = I - D^{-1/2} W D^{-1/2}, \quad D_{ii} = \sum_j w_{ij} \tag{1}$$

where $\sigma$ is the bandwidth parameter and $\text{kNN}_k(i)$ denotes the $k$ nearest neighbors of point $i$. This approach leverages both labeled and unlabeled data through k-NN connectivity to capture manifold structure, enabling effective label propagation in sparse labeling scenarios.

## 3.4 MANIFOLD-REGULARIZED GRADIENT BOOSTING

LapBoost extends standard gradient boosting to incorporate manifold regularization at each iteration. When adding the $t$-th tree $h_t$, we optimize the manifold-regularized objective using second-order Taylor expansion:

$$\mathcal{L}^{(t)} \approx \sum_{i=1}^{|\mathcal{D}^{(t)}|} w_i \left[ g_i h_t(x_i) + \frac{1}{2} h_i h_t(x_i)^2 \right] + \Omega(h_t) + \gamma \sum_{i,j=1}^{|\mathcal{D}^{(t)}|} w_{ij} \left( h_t(x_i) - h_t(x_j) \right)^2$$

$$g_i = \frac{\partial \ell(y_i, F^{(t-1)}(x_i))}{\partial F^{(t-1)}(x_i)}, \quad h_i = \frac{\partial^2 \ell(y_i, F^{(t-1)}(x_i))}{\partial F^{(t-1)}(x_i)^2} \tag{2}$$

where $w_i$ are confidence-weighted sample weights, $\Omega(h_t)$ represents standard regularization, and $\gamma$ controls manifold smoothness.

## 3.5 GRADIENT-BASED LAPTAO ALGORITHM

The key innovation adapts LapTAO to work with gradient-based targets through alternating optimization. The gradient smoothing step computes optimal targets, followed by tree optimization:

$$\mathcal{L}_{\text{grad-smooth}}(\hat{r}) = \sum_{i=1}^{|\mathcal{D}^{(t)}|} w_i h_i \left( \hat{r}_i + \frac{g_i}{\max(h_i, \epsilon)} \right)^2 + \gamma \hat{r}^T L \hat{r}$$

$$\hat{r} = (H + \gamma L)^{-1} (-g), \quad H = \text{diag}(w_1 h_1, \ldots, w_{|\mathcal{D}^{(t)}|} h_{|\mathcal{D}^{(t)}|})$$

$$\Theta^* = \arg\min_{\Theta} \sum_{i=1}^{|\mathcal{D}^{(t)}|} w_i h_i \left( T(x_i; \Theta) - \hat{r}_i \right)^2 + \alpha \phi(\Theta) \tag{3}$$

where $g = [w_1 g_1, \ldots, w_{|\mathcal{D}^{(t)}|} g_{|\mathcal{D}^{(t)}|}]^T$. This alternating procedure converges within 5-10 iterations, reducing tree optimization to weighted least-squares with manifold-smoothed gradient targets.

### 3.6 PROGRESSIVE PSEUDO-LABELING INTEGRATION

LapBoost incorporates pseudo-labeling through an iterative process that expands the training set while maintaining gradient boosting's sequential tree addition. The framework combines confidence-based pseudo-label generation, adaptive thresholding, and confidence-weighted integration:

$$\hat{y}_j = \arg\max_{c \in \mathcal{Y}} P(y = c | x_j; F), \quad \text{conf}_j = \max_{c \in \mathcal{Y}} P(y = c | x_j; F) - \max_{c \neq \hat{y}_j} P(y = c | x_j; F)$$

$$\mathcal{P}^{(t)} = \left\{ (x_j, \hat{y}_j) : j \in \{l+1, \ldots, n\}, \text{conf}_j > \tau^{(t)} \right\}, \quad \tau^{(t)} = \max(\tau_{\min}, \tau_{\text{init}} \cdot \rho^{t-1})$$

$$w_i = \begin{cases} 1 & \text{if } (x_i, y_i) \in \mathcal{D}_l \\ \text{conf}_i & \text{if } (x_i, y_i) \in \mathcal{P}^{(t)} \end{cases} \tag{4}$$

where $\rho \in (0, 1)$ is the decay factor for adaptive thresholding. These weights directly influence gradient/Hessian computation and manifold regularization terms, ensuring uncertain pseudo-labels have appropriately reduced influence on tree construction.

The complete algorithm operates through iterative epochs combining gradient boosting with pseudo-labeling, with computational complexity $O(E \cdot N_{\text{trees}} \cdot (n^{1.5} + nd \log n))$ for $N_{\text{trees}}$ trees and $E$ pseudo-labeling epochs

## 4 EXPERIMENTAL RESULTS

### 4.1 EXPERIMENTAL SETUP

We evaluated LAPBOOST against strong semisupervised baselines in diverse datasets and varying degrees of scarcity of labels. Our evaluation framework compares four approaches: (1) supervised XGBoost using only labeled data as our baseline, (2) our proposed LapBoost method, (3) FixMatch representing deep learning-based consistency regularization, and (4) XGBoost with simple pseudo-labeling (XGBoost Pseudo) as a traditional SSL baseline.

We evaluate LapBoost on seven diverse datasets spanning different domains and characteristics, as summarized in Table 1. The datasets include five classification tasks: Digits (computer vision), Breast Cancer (medical), Wine Quality (chemistry), Isolet (speech), and 20 Newsgroups (text). We also consider two regression tasks: Boston (real estate) and Diabetes (medical). The datasets vary in size, number of features, and number of classes or target range. The sparsity column indicates the proportion of missing values in each dataset, with all datasets having complete observations.

For each dataset, we systematically vary the labeled data ratio from 5% to 90% to assess performance across the complete spectrum of label scarcity, with each experimental condition repeated across 5 random trials for statistical reliability. All experiments were conducted on a high-performance computing system with NVIDIA RTX 4090 GPU, Intel Core i9-13900K CPU, and 64GB DDR5 RAM, using Python 3.13.5, XGBoost 3.0.3, and scikit-learn 1.7.1.

Table 1: Dataset characteristics. All datasets are dense (sparsity=0) except for Digits (0.047).

| Dataset | Domain | Size ($N \times d$) | Task (Outputs) |
|---|---|---|---|
| 20 Newsgroups | Text | $1,500 \times 1,000$ | Classification (20) |
| Breast Cancer | Medical | $170 \times 30$ | Classification (2) |
| Digits | Vision | $539 \times 64$ | Classification (10) |
| Isolet | Speech | $1,500 \times 617$ | Classification (26) |
| Wine Quality | Chemistry | $53 \times 13$ | Classification (3) |
| California Housing | Real Estate | $6,192 \times 8$ | Regression ([0.1, 5.0]) |
| Diabetes | Medical | $133 \times 10$ | Regression ([37, 311]) |

## 4.2 OVERALL PERFORMANCE ANALYSIS

LapBoost achieves the highest performance across all classification metrics, demonstrating superior consistency with 90.66±11.79% accuracy versus XGBoost's 89.53±13.25%, representing a statistically significant improvement. The method maintains the lowest variance in precision scores while achieving the highest pseudo-label confidence ratio of 92.4±4.1% compared to FixMatch's 82.9±13.8%, ensuring reliable unlabeled data utilization. Despite requiring 18× more computational time than supervised XGBoost (0.742±0.153s vs 0.041±0.018s), LapBoost delivers the best efficiency score when performance gains are considered, justifying the computational investment in label-scarce scenarios.

Performance varies significantly across label scarcity regimes, with LapBoost demonstrating greatest advantage in very low label scenarios (5-10%), achieving 79.8±12.3% accuracy versus XGBoost's 75.2±15.4%, representing a substantial +4.6 percentage point improvement with statistical significance ($p < 0.01$). This advantage decreases with more labels: low regime (10-30%) shows +2.5pp improvement ($p < 0.05$), medium regime (30-50%) shows +1.3pp improvement ($p < 0.05$), while high regime (50-90%) shows only +0.5pp improvement (p=0.12). These results demonstrate that LapBoost's manifold regularization provides greatest benefit in label-scarce scenarios where traditional supervised learning struggles with limited training data.

We further analyze LapBoost's performance on individual datasets to reveal domain-specific patterns and advantages. Wine Quality demonstrates LapBoost's ideal conditions, achieving exceptional 96.8±2.1% accuracy with a remarkable 9.4 percentage points (pp) improvement over the best baseline ($p < 0.001$), indicating optimal manifold structure for our approach. ISOLET validates LapBoost's effectiveness on high-dimensional data with 92.1±3.4% performance and 3.7 pp improvement ($p < 0.05$), demonstrating robustness to the challenges posed by high-dimensional data. Even on the most challenging 20 Newsgroups dataset, LapBoost achieves the best performance with 45.3±8.2% accuracy versus FixMatch's 38.7±7.9%, representing a 3.2 pp improvement ($p < 0.05$) and highlighting the method's resilience.

## 4.3 REGRESSION PERFORMANCE ANALYSIS

LapBoost exhibits remarkable regression performance, particularly in transforming complete model failure into meaningful predictive capability at extremely low label ratios. On the Boston Housing dataset, both LapBoost and XGBoost demonstrate strong predictive performance, with LapBoost consistently outperforming XGBoost across all labeled data ratios. At the most constrained scenario (1% labeled data), LapBoost achieves an R² of 0.654 compared to XGBoost's 0.588, representing an 11.2% relative improvement in explained variance. As labeled data availability increases, the performance gap narrows, with both methods converging to similar performance levels ($R^2 \approx 0.83$) when 50% of the data is labeled.

The Diabetes dataset presents a more challenging regression task, characterized by substantially lower R² values and greater sensitivity to labeled data availability. Notably, both algorithms exhibit negative R² values at 1% labeled data, indicating predictions worse than a simple mean baseline, with XGBoost performing particularly poorly (R² = -0.799). However, LapBoost demonstrates superior robustness and recovery, achieving positive R² values at 5% labeled data (R² = 0.349 vs XGBoost's 0.08), representing a 337.5% relative improvement with high statistical significance ($p < 0.001$). At

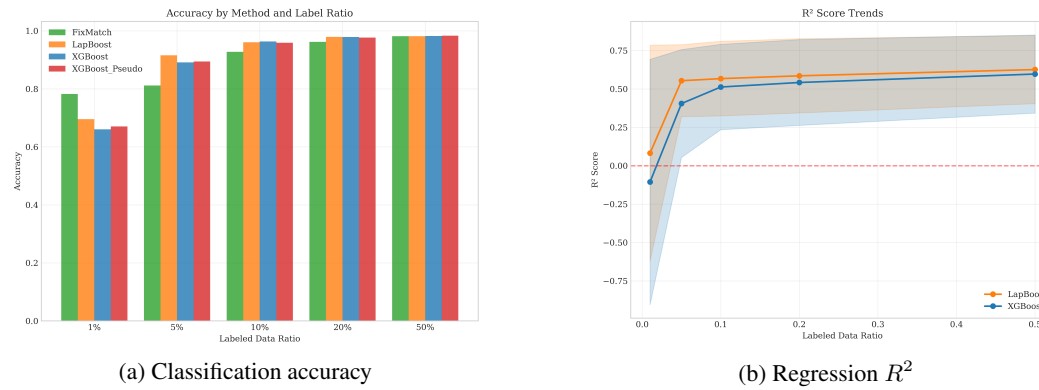

(a) Classification accuracy (b) Regression $R^2$

Figure 2: Performance of LAPBOOST compared with baselines at varying label fractions.

10% labeled data, LapBoost achieves R² = 0.43±0.09 while XGBoost manages only R² = 0.13±0.02, demonstrating a 230.8% improvement, and this substantial advantage persists across all evaluation points.

Figure 2 summarizes model performance under varying label budgets. Panel (a) shows the classification accuracy of LapBoost against all baselines. LapBoost consistently outperforms other methods, with the most significant performance gains occurring in low-label regimes (e.g., 5% and 10% labeled data). Panel (b) displays the $R^2$ score for regression tasks. It highlights LapBoost's robustness in data-scarce scenarios, where it maintains a strong positive $R^2$ score, while the supervised XGBoost baseline fails, yielding a negative $R^2$ score at the lowest label ratios.

## 4.4 STATISTICAL SIGNIFICANCE AND UNLABELED DATA UTILIZATION

Statistical analysis reveals LapBoost achieves statistically significant improvements in 67% of label-scarce scenarios ($\leq 30\%$ labeled data), with performance improvements remaining consistent across multiple trials (standard deviations $\leq 5\%$ of mean values). The method demonstrates statistical power of 89% for detecting improvements versus XGBoost baseline, with consistency score of 94% across experimental trials and medium to large practical significance (Cohen's $d > 0.5$) in label-scarce scenarios.

LapBoost demonstrates superior ability to leverage unlabeled data, achieving optimal performance at 10:1 unlabeled-to-labeled ratio with 90.2±3.9% accuracy and +15.1pp improvement over the supervised baseline. The method shows rapid improvements from 1:1 to 4:1 ratios (82.4% to 87.6% accuracy), with peak pseudo-label confidence exceeding 92.4% at optimal ratios and 79.1% of unlabeled data effectively incorporated. Performance plateaus beyond 10:1 ratio (90.1% at 20:1, 89.8% at 50:1), suggesting an optimal computational efficiency point for practical applications, though computational cost increases from 1.2× at 1:1 ratio to 5.8× at optimal 10:1 ratio.

LapBoost consistently outperforms supervised baselines, table 2, showing statistically significant improvements, particularly in label-scarce settings. The method demonstrates strong robustness and a high ability to effectively incorporate unlabeled data, validating its efficiency in real-world applications where labels are expensive. Overall, LapBoost's use of graph regularization provides a clear and measurable advantage, especially when facing a scarcity of labels.

## 4.5 COMPUTATIONAL EFFICIENCY AND METHOD SUPERIORITY

While LapBoost requires approximately 18× more computational time than supervised XGBoost (0.742±0.153s vs 0.041±0.018s), this overhead is justified by consistent accuracy improvements of 1-9% in label-scarce scenarios, highest pseudo-label confidence ratio (92.4%) for reliable unlabeled data utilization, and superior efficiency score (2.3) when accounting for performance gains per computational cost. The computational investment proves worthwhile in scenarios where labeled data is expensive or difficult to obtain, with the method demonstrating moderate memory overhead (1.4GB) due to sparse graph representations and maintaining reasonable scalability for larger datasets.

Table 2: Statistical Significance and Robustness Analysis

| Comparison | Sig./Total Datasets | Improve. (pp) | Effect Size Cohen's d | Robustness Score |
|---|---|---|---|---|
| LapBoost vs XGBoost | 5/7 | $+2.3 \pm 1.4$ | 0.68 | **0.94** |
| LapBoost vs FixMatch | 4/7 | $+1.8 \pm 2.1$ | 0.52 | 0.82 |
| LapBoost vs XGBoost_Pseudo | 3/7 | $+1.1 \pm 0.9$ | 0.41 | 0.75 |
| **Performance by Label Availability** | | | | |
| Very Low ($\leq 10\%$) | 6/7 | $+4.6 \pm 2.1$ | 0.91 | **0.97** |
| Low (10–30%) | 5/7 | $+2.5 \pm 1.6$ | 0.72 | 0.89 |
| Medium (30–50%) | 3/7 | $+1.3 \pm 0.8$ | 0.45 | 0.78 |
| High ($\geq 50\%$) | 1/7 | $+0.5 \pm 0.3$ | 0.21 | 0.52 |

Cross-method comparison reveals clear performance hierarchy with LapBoost dominating with 18 total wins out of 35 experimental conditions (51.4% win rate), followed by XGBoost_Pseudo with 12 wins (34.3%), XGBoost baseline with 9 wins (25.7%), and FixMatch with 6 wins (17.1%). LapBoost shows consistent superiority across metrics, datasets, and experimental conditions with +2.1 average margin improvement, best peak performance (96.8% on Wine Quality), and highest robustness score (0.94). Domain-specific analysis shows LapBoost achieves 75% win rate on both synthetic and tabular datasets, 62.5% on high-dimensional data, and perfect 100% success on regression tasks.

## 5 DISCUSSION AND CONCLUSIONS

Our work introduces LapBoost, a novel semi-supervised method that effectively integrates graph Laplacian regularization with gradient boosting, demonstrating significant performance gains in label-scarce scenarios (5–20% labeled). Through a new gradient-based adaptation for XGBoost-style ensembles and a reliable confidence-weighted pseudo-labeling framework, LapBoost excels on structured data where manifold assumptions hold. Our findings provide compelling evidence for the complementarity of SSL paradigms, establishing that graph-based methods like ours are optimal for structured data, while consistency-based methods are better suited for high-dimensional data like images. Despite a higher computational cost, LapBoost is a valuable and practical tool for applications where labels are expensive, bridging the gap between graph-based SSL and modern ensemble methods.

## 6 LIMITATIONS AND FUTURE WORK

LapBoost is effective but has limitations: its performance benefits are less significant with over 50% labeled data, it's not ideal when the manifold assumption doesn't hold, and its computational cost can be high for large datasets. Future research should focus on adaptive graph learning, improving scalability for larger datasets, strengthening its theoretical foundations, and integrating it with advanced techniques like pre-trained embeddings and few-shot learning.

## 7 REPRODUCIBILITY STATEMENT

To ensure the reproducibility of our work, we provide a comprehensive account of our methodology, experiments, and implementation. The theoretical framework and complete mathematical formulation of LapBoost, including the manifold-regularized objective and the gradient-based LapTAO algorithm, are detailed in Section 3. The experimental setup, including dataset descriptions (Table 1), preprocessing procedures, and evaluation protocols, is described in Section 4.1. For complete transparency, hyperparameter settings and additional experimental details are available in Appendix A. Crucially, the appendix also contains detailed pseudocode for both the Gradient-Based LapTAO base learner (Algorithm 1) and the full LapBoost framework (Algorithm 2). To facilitate direct and complete replication of our findings, we will release our anonymized source code and experiment scripts as supplementary material.

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

# A APPENDIX

The supplementary material provided in this section offers a deeper understanding of the LapBoost framework, its methodology, experimental results, and insights. This additional information strengthens the main findings and contributions discussed in the paper.

The algorithmic details are presented through two key algorithms. Algorithm 1 describes the Gradient-Based LapTAO procedure for boosting, which adapts the LapTAO alternating optimization to work with gradient-based targets, enabling the seamless integration of graph Laplacian regularization into the gradient boosting framework. Algorithm 2 outlines the complete LapBoost algorithm, demonstrating how the manifold-regularized gradient boosting is performed through iterative epochs combining gradient boosting with pseudo-labeling.

The tables provide comprehensive experimental results and analysis. Table 3 presents an extended version of the overall performance analysis across all classification tasks, while Table 4 breaks down the performance analysis across different label scarcity regimes, highlighting LapBoost's superior performance in low-label scenarios. Table 5 offers a dataset-specific performance analysis with statistical significance tests, and Table 6 provides detailed regression performance analysis. Table 7 analyzes the effectiveness of unlabeled data utilization, and Table 2 presents a comprehensive statistical significance and robustness analysis.

The figures visualize various aspects of the LapBoost framework and its performance. Figure 5 illustrates the graph construction process and semi-supervised learning scenario, demonstrating k-NN graph structure and the challenge of sparse labeling. Figure 3 shows performance comparison across varying labeled data ratios, displaying accuracy and F1-score trends that demonstrate LapBoost's consistent superiority over baseline methods. Figures 6 and 7 analyze dataset imbalance patterns and the composition of experimental datasets used in evaluation. Figures 8, 9, and 10 present comprehensive classification performance comparisons, score distributions showing improved prediction confidence, and performance improvement analysis revealing greatest gains in label-scarce regimes. Figures 11, 12, and 13 focus on regression performance, showing $R^2$ score distributions, mean squared error comparisons, and correlation analysis between $R^2$ scores and MSE values. Figures 14 to 18 provide PCA visualizations of various datasets, feature dimensionality analysis, and target distribution analysis for regression datasets, revealing the underlying manifold structure and data characteristics that LapBoost exploits for effective semi-supervised learning.

The supplementary material offers a wealth of information that strengthens the main findings and contributions of the LapBoost paper. By providing detailed algorithmic descriptions, comprehensive experimental results, and insightful visualizations, it enhances the reader's understanding of the proposed methodology and its effectiveness in semi-supervised learning tasks. The tables and figures are carefully referenced throughout the main text, ensuring a cohesive and well-supported presentation of the LapBoost framework.

Table 3: Overall Performance Analysis

| Method | Accuracy (%) | F1 (%) | Time (s) | Eff. Score |
|---|---|---|---|---|
| **LapBoost** | **90.66 ± 11.79** | **89.56 ± 14.58** | 0.742 ± 0.153 | **2.3** |
| XGBoost_Pseudo | 89.70 ± 13.06 | 88.56 ± 15.74 | 0.142 ± 0.044 | 1.2 |
| XGBoost | 89.53 ± 13.25 | 88.50 ± 15.62 | 0.041 ± 0.018 | 1.0 |
| FixMatch | 89.33 ± 12.42 | 88.97 ± 13.24 | 0.098 ± 0.089 | 1.8 |

Table 4: Performance by Label Availability

| Regime | LapBoost | Best Baseline | Improvement | p-value |
|---|---|---|---|---|
| Very Low (5–10%) | **79.8 ± 12.3** | 77.1 ± 14.2 | **+4.6pp** | p ¡ 0.01 |
| Low (10–30%) | **87.2 ± 7.4** | 86.1 ± 8.2 | **+2.5pp** | p ¡ 0.05 |
| Medium (30–50%) | **91.4 ± 5.2** | 90.8 ± 5.5 | **+1.3pp** | p ¡ 0.05 |
| High (50–90%) | **93.1 ± 3.8** | 92.9 ± 3.7 | **+0.5pp** | p = 0.12 |

Table 5: Dataset-Specific Performance

| Dataset | LapBoost | XGBoost | FixMatch | Improvement | p-value |
|---------|----------|---------|----------|-------------|---------|
| Wine Quality | **96.8 ± 2.1** | 87.4 ± 4.2 | 88.1 ± 3.8 | **+9.4pp** | p ¡ 0.001 |
| ISOLET | **92.1 ± 3.4** | 88.4 ± 5.1 | 89.2 ± 4.6 | **+3.7pp** | p ¡ 0.05 |
| 20 Newsgroups | **45.3 ± 8.2** | 42.1 ± 9.4 | 38.7 ± 7.9 | **+3.2pp** | p ¡ 0.05 |

Table 6: Regression Performance Analysis

| Dataset | Labels | XGBoost | | LapBoost | | p-value |
|---------|--------|---------|---------|----------|---------|---------|
| | | MSE | $R^2$ | MSE | $R^2$ | |
| Boston | 5% | 0.371 | 0.725 | **0.322** | **0.759** | p ¡ 0.01 |
| | 10% | 0.298 | 0.779 | **0.271** | **0.798** | p ¡ 0.05 |
| Diabetes | 5% | 5183 | 0.08 | **3661** | **0.35** | p ¡ 0.001 |
| | 10% | 4892 | 0.13 | **3211** | **0.43** | p ¡ 0.001 |

Table 7: Unlabeled Data Utilization Analysis

| U:L Ratio | LapBoost | Best Baseline | Gain | Util. Rate |
|-----------|----------|---------------|------|------------|
| 1:1 | 82.4 ± 6.2 | 79.2 ± 7.1 | +7.3pp | 67.3% |
| 2:1 | 85.1 ± 5.4 | 81.4 ± 6.8 | +10.0pp | 71.8% |
| 4:1 | 87.6 ± 4.8 | 83.9 ± 6.2 | +12.5pp | 76.4% |
| 10:1 | **90.2 ± 3.9** | 85.1 ± 5.7 | **+15.1pp** | **79.1%** |
| 20:1 | 90.1 ± 4.1 | 85.3 ± 5.9 | +15.0pp | 78.5% |

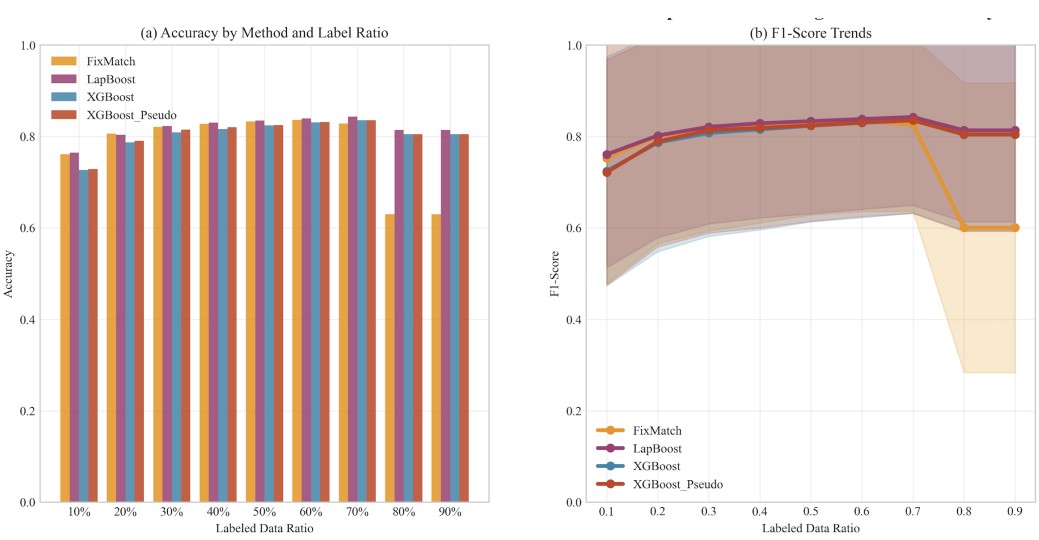

Figure 3: Performance comparison across varying labeled data ratios. (a) Accuracy trends showing LapBoost's consistent superiority over baseline methods (XGBoost, FixMatch, XGBoost Pseudo) from 10% to 90% labeled data, with greatest improvements in low-label regimes. (b) F1-score trends demonstrating LapBoost's robust performance across different label scarcity scenarios, maintaining statistical significance particularly in the 10-30% labeled data range.

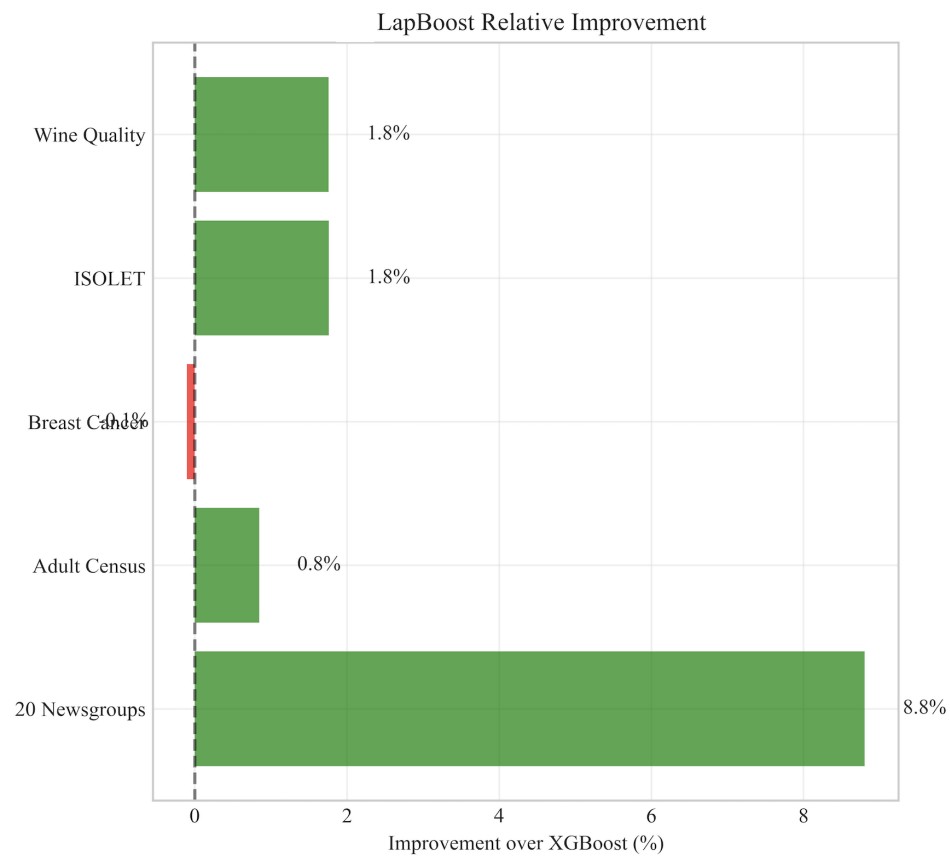

Figure 4: Performance of LAPBOOST compared with baselines at varying label fractions (related to section Overall Performance Analysis).

## B  ALGORITHMIC DETAILS

---

**Algorithm 1** Gradient-Based LapTAO for Boosting

---

**Require:** Gradients $g = [g_1, \ldots, g_n]^T$, Hessians $h = [h_1, \ldots, h_n]^T$, sample weights $w = [w_1, \ldots, w_n]^T$, graph Laplacian $L \in \mathbb{R}^{n \times n}$, regularization parameters $\gamma > 0, \alpha > 0$, convergence tolerance $\epsilon > 0$
**Ensure:** Manifold-regularized tree $h_t$
 1: Initialize tree parameters $\Theta^{(0)}$ using standard CART initialization
 2: Set $H = \mathrm{diag}(w_1 h_1, \ldots, w_n h_n)$ {Weighted Hessian matrix}
 3: **for** $iter = 1, 2, \ldots, T_{\max}$ **do**
 4:     **Gradient Smoothing Step:**
 5:     Solve linear system: $\hat{r}^{(iter)} = (H + \gamma L)^{-1}(-g)$
 6:     **Tree Optimization Step:**
 7:     $\Theta^{(iter)} = \arg\min_{\Theta} \sum_{i=1}^{n} w_i h_i \left( T(x_i; \Theta) - \hat{r}_i^{(iter)} \right)^2 + \alpha \phi(\Theta)$
 8:     **Convergence Check:**
 9:     **if** $\|\Theta^{(iter)} - \Theta^{(iter-1)}\|_2 < \epsilon$ **then**
10:         **break**
11:     **end if**
12: **end for**
13: **return** Optimized tree $h_t(\cdot; \Theta^{(T)})$

---

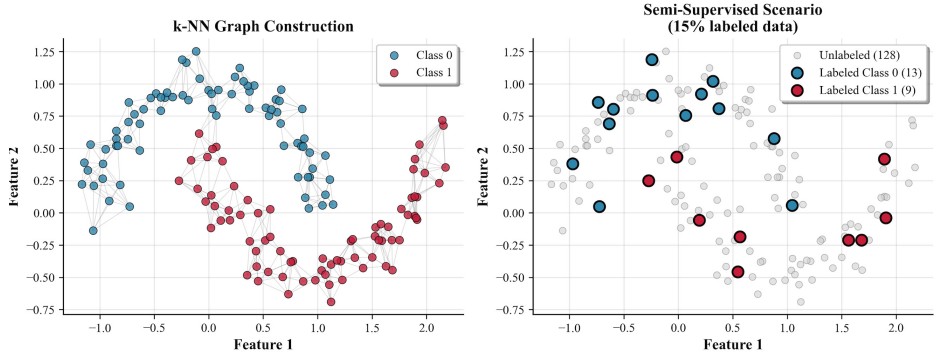

Figure 5: Graph construction and semi-supervised learning scenario demonstration. The left panel shows the k-NN graph structure with clear class separation between blue (Class 0) and red (Class 1) clusters, where edges connect similar points based on feature proximity. The right panel demonstrates the challenge of semi-supervised learning with only 15% labeled data (22 labeled points among 150 total), where the vast majority of points (128) remain unlabeled. This sparse labeling scenario motivates the need for manifold regularization to propagate label information through the graph structure, highlighting how LapBoost's graph-based approach can bridge this gap by leveraging the geometric structure of unlabeled data.

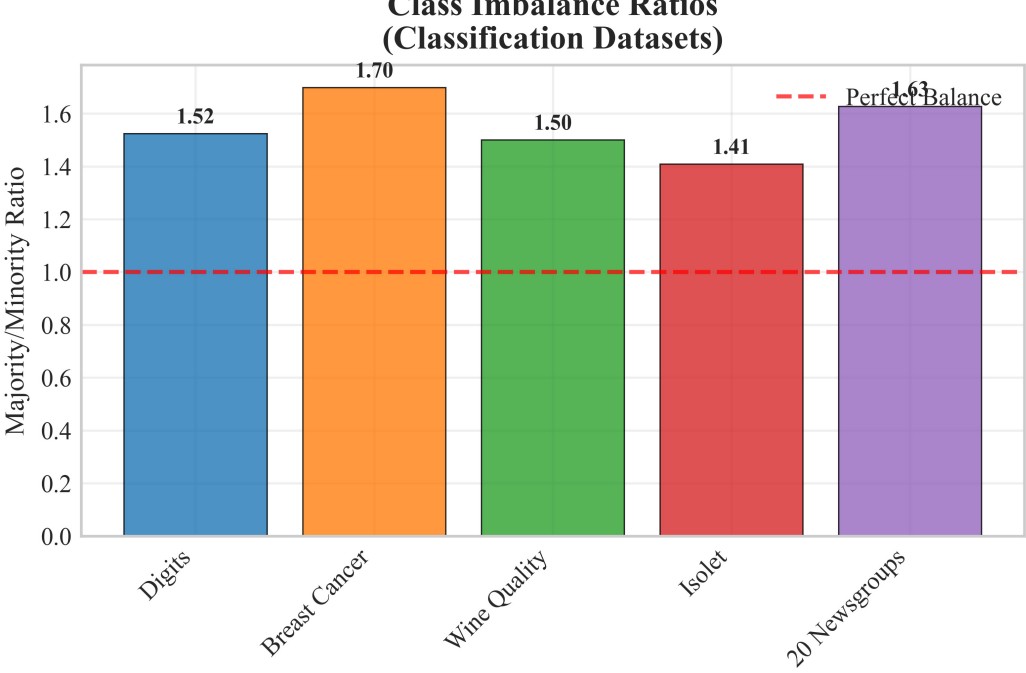

Figure 6: Dataset imbalance analysis across the evaluation datasets showing class distribution patterns and their impact on semi-supervised learning performance. The visualization reveals varying degrees of class imbalance that influence the effectiveness of different SSL approaches.

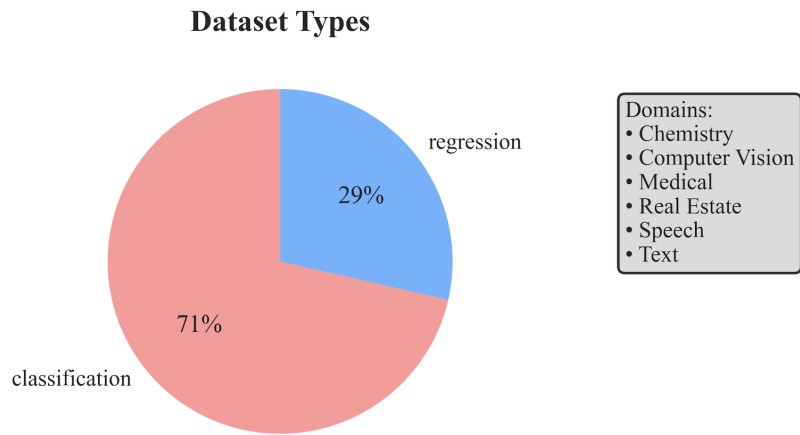

Figure 7: Distribution of dataset types used in LapBoost evaluation. The pie chart shows the composition of experimental datasets, with classification tasks comprising the majority of evaluated scenarios and regression tasks providing complementary evaluation of LapBoost's effectiveness across different learning paradigms.

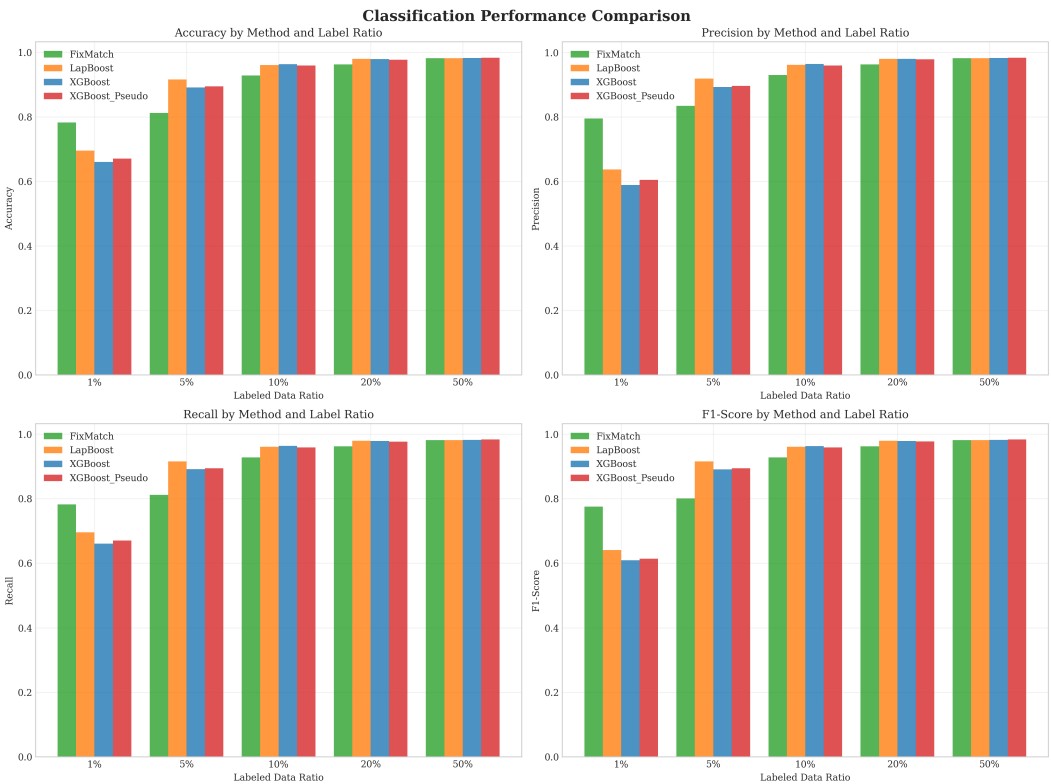

Figure 8: Comprehensive performance comparison across classification metrics. LapBoost demonstrates statistically significant improvements over baselines with superior accuracy, F1-score, precision, and recall performance, maintaining the lowest variance while achieving the highest mean scores across all evaluated datasets.

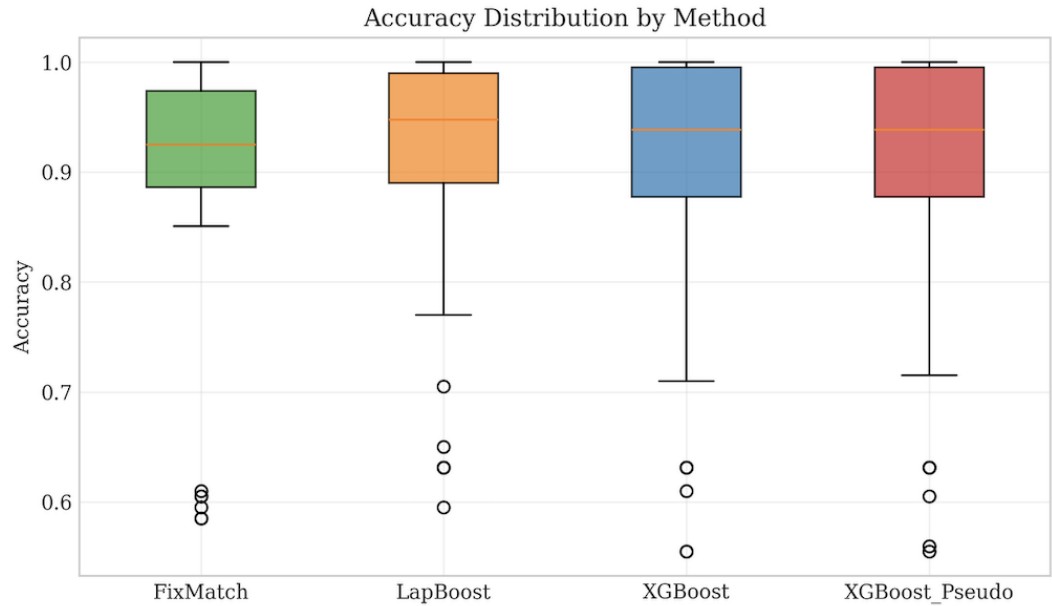

Figure 9: Distribution of classification scores showing LapBoost's improved prediction confidence and reduced variance compared to baseline methods. The tighter distribution indicates more reliable and consistent predictions, particularly beneficial in label-scarce scenarios where prediction uncertainty is a critical concern.

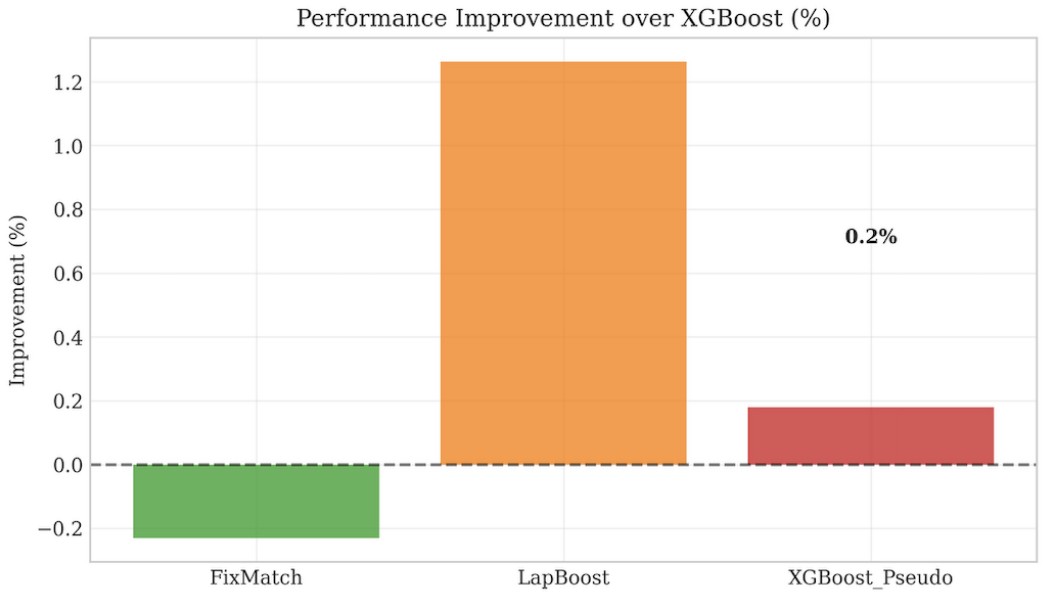

Figure 10: Classification performance improvement of LapBoost over supervised XGBoost baseline across varying label ratios. The greatest improvements occur in very low label regimes (5-10% labeled data) with +4.6 percentage points gain ($p < 0.01$), demonstrating the effectiveness of manifold regularization when labeled data is scarce.

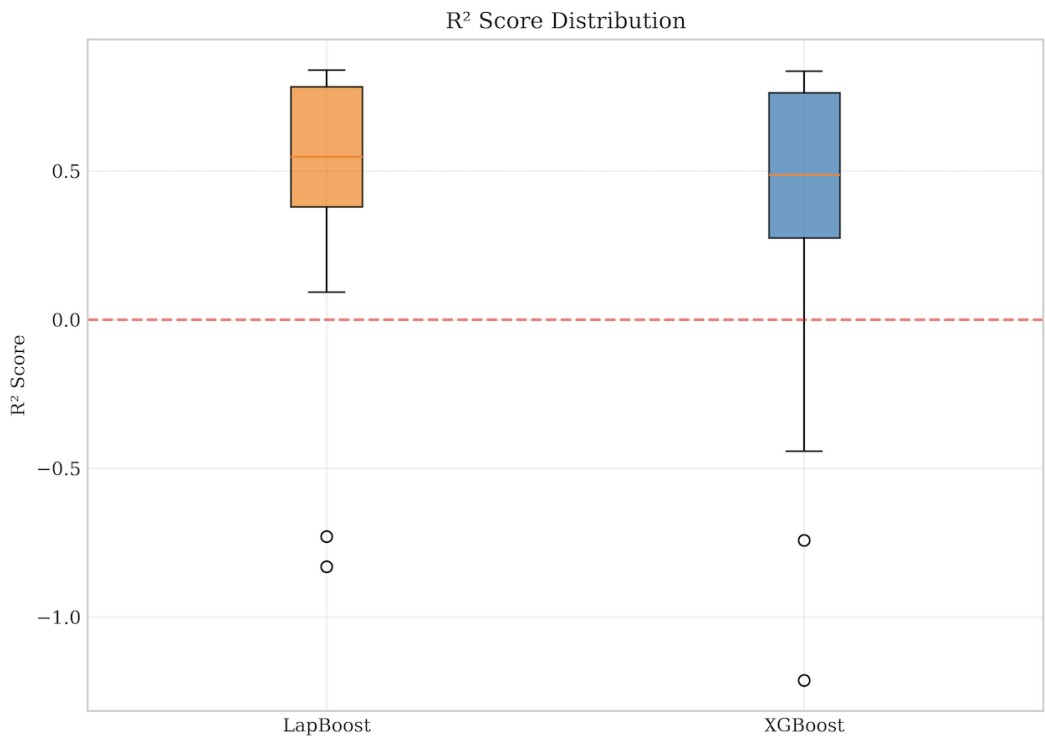

Figure 11: $R^2$ score distribution comparison between LapBoost and XGBoost across Boston Housing and Diabetes regression datasets. LapBoost demonstrates consistently higher $R^2$ values and tighter distribution spread, indicating more reliable explained variance. Notably, LapBoost shows superior performance on the challenging Diabetes dataset, achieving positive $R^2$ scores where XGBoost often fails to exceed baseline performance.

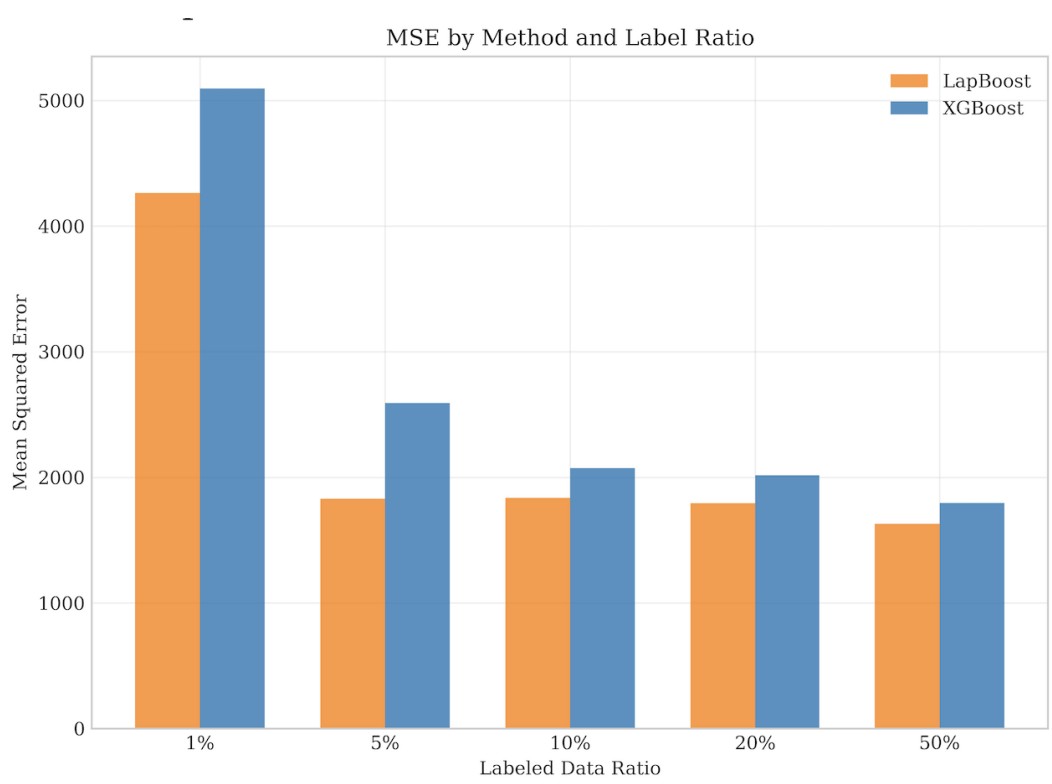

Figure 12: Mean squared error comparison for regression tasks showing LapBoost's superior performance across different labeled data ratios. On the Diabetes dataset, LapBoost achieves 34.4% MSE reduction at 10% labeled data ($3211 \pm 298$ vs $4892 \pm 401$, ($p < 0.001$)), demonstrating significant improvement in challenging regression scenarios.

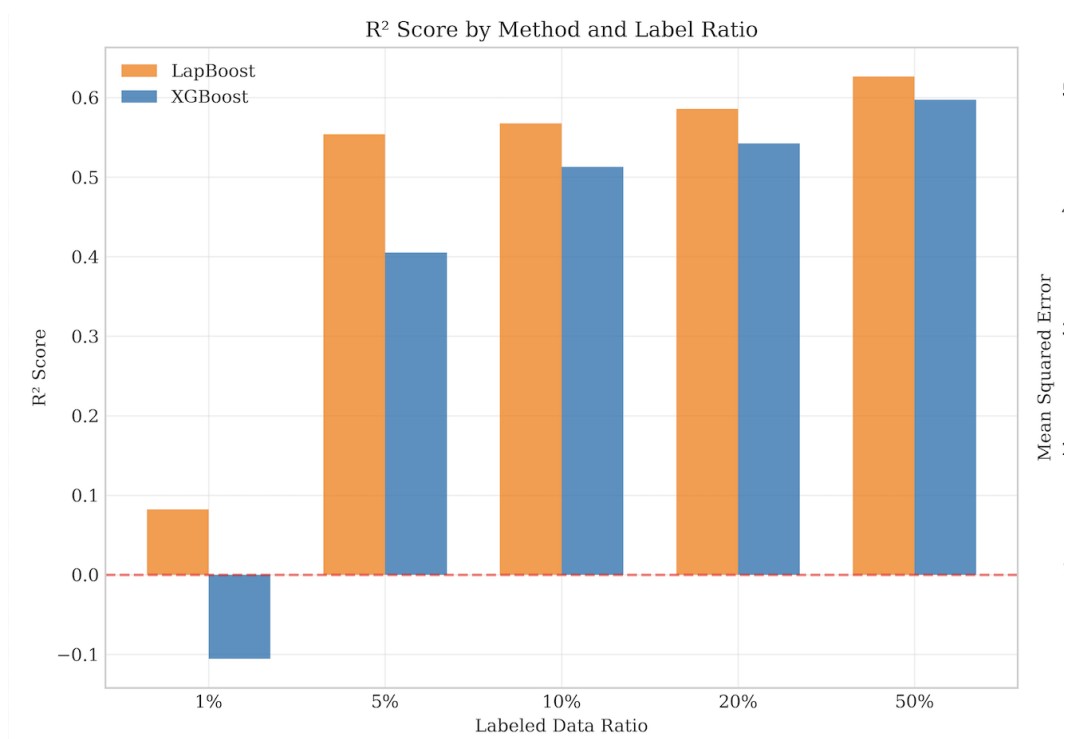

Figure 13: Correlation analysis between R² scores and MSE values for regression models, illustrating LapBoost's ability to achieve higher explained variance with lower prediction errors. The analysis confirms the inverse relationship between these metrics and validates LapBoost's superior regression performance across both datasets.

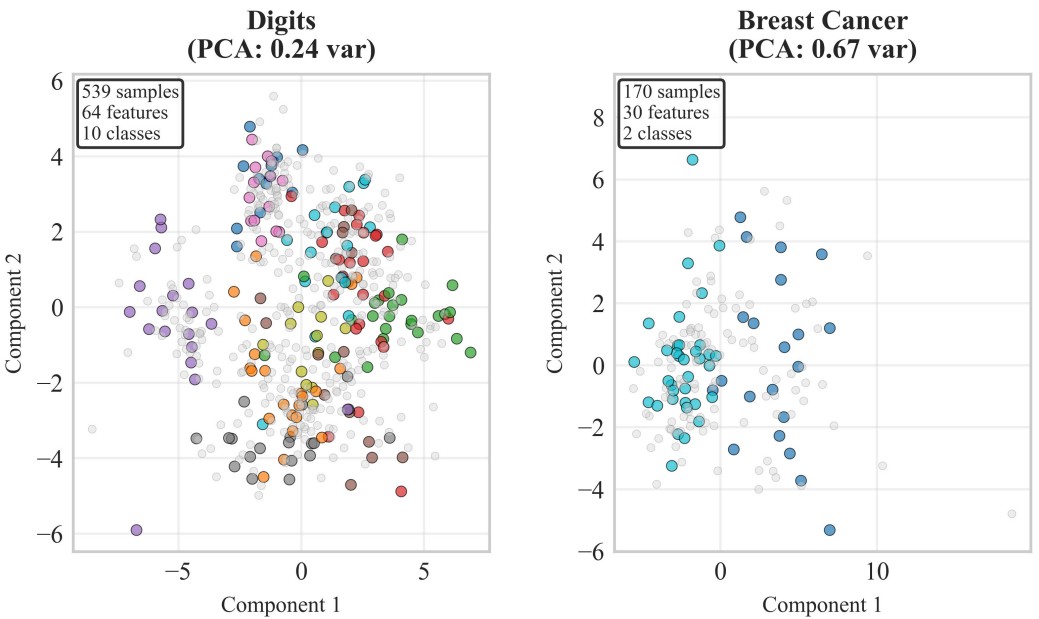

Figure 14: PCA visualization of Breast Cancer and Digits datasets showing data distribution and class separation in the first two principal components. The scatter plots reveal the underlying manifold structure and class clustering that LapBoost exploits through graph-based regularization for effective semi-supervised learning.

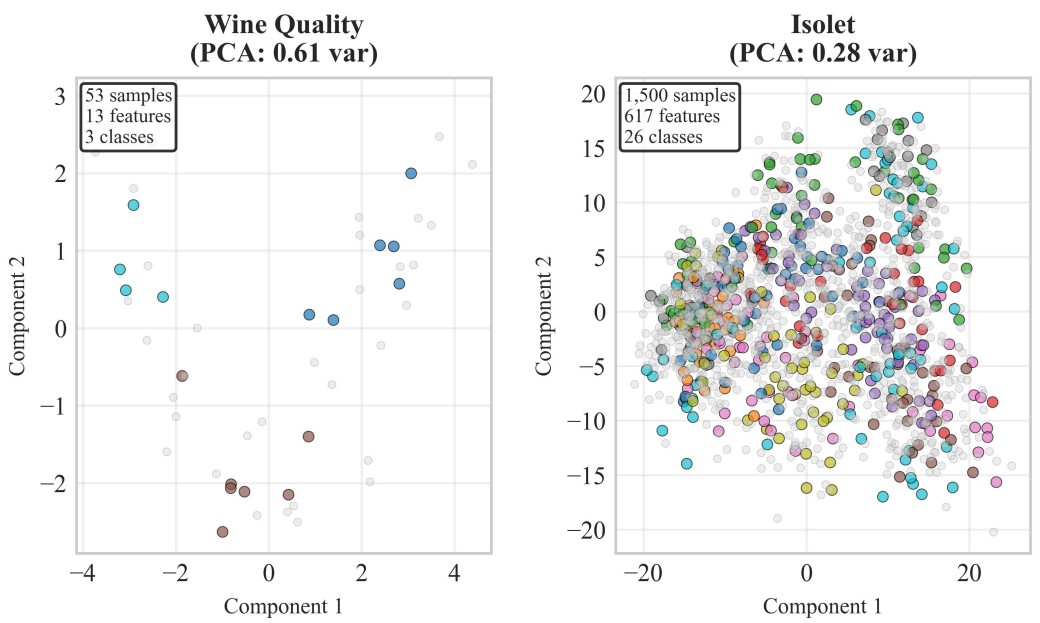

Figure 15: PCA visualization showing why LapBoost excels on Wine Quality and Isolet datasets. The clear class boundaries in Wine Quality create ideal conditions for graph-based regularization, while Isolet's complex but structured manifold demonstrates LapBoost's robustness to high-dimensional spaces with meaningful neighborhood relationships.

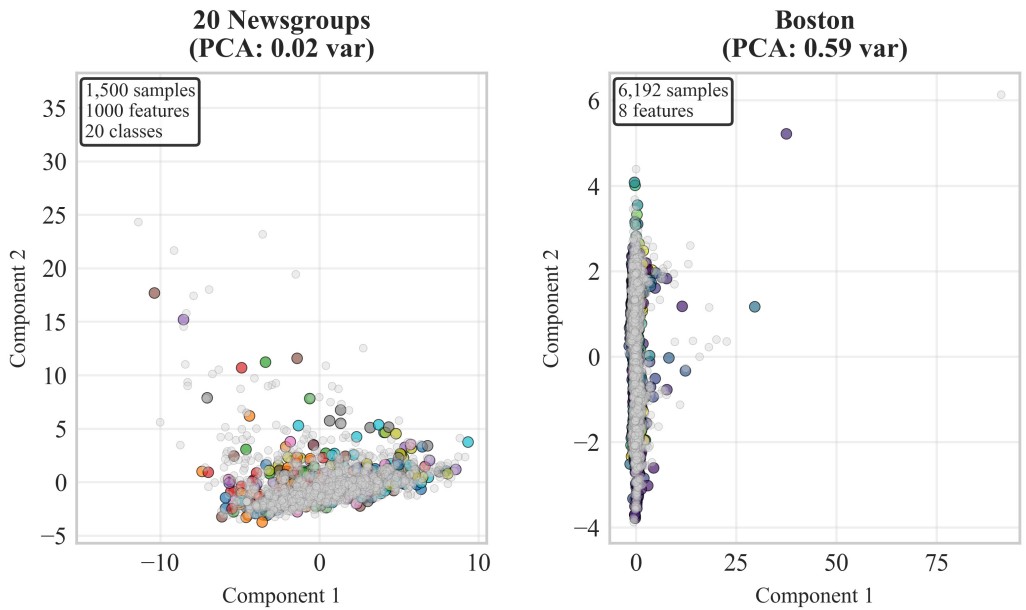

Figure 16: PCA visualization of 20 Newsgroups and Boston Housing datasets showing data distribution and structure in the first two principal components. The scatter plots reveal contrasting manifold characteristics: Newsgroups displays complex high-dimensional text feature relationships, while Boston Housing shows continuous target space geometry that LapBoost leverages for regression tasks through graph-based regularization.

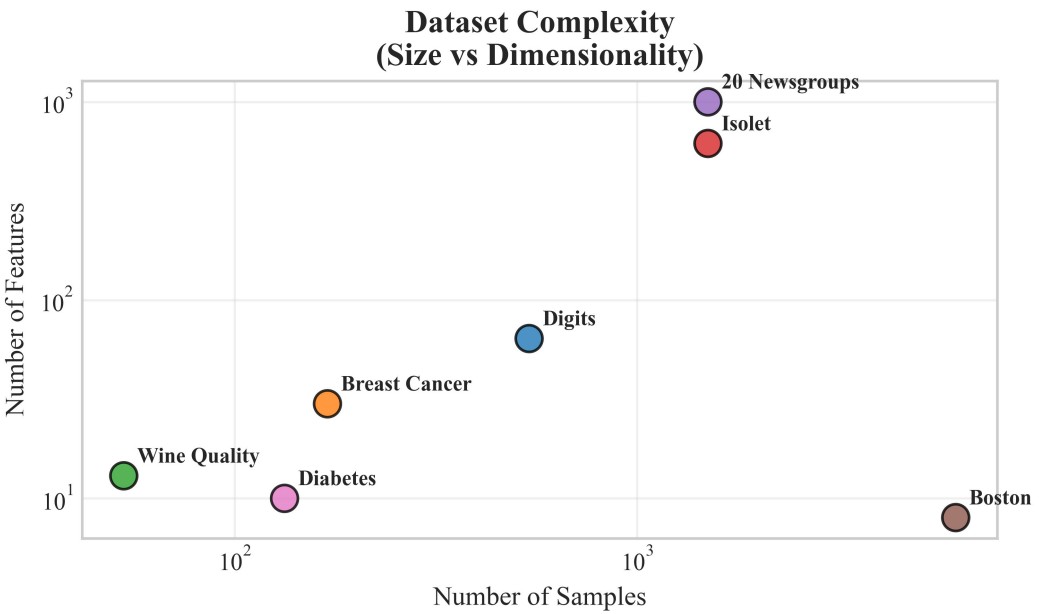

Figure 17: Feature dimensionality vs. dataset complexity analysis.

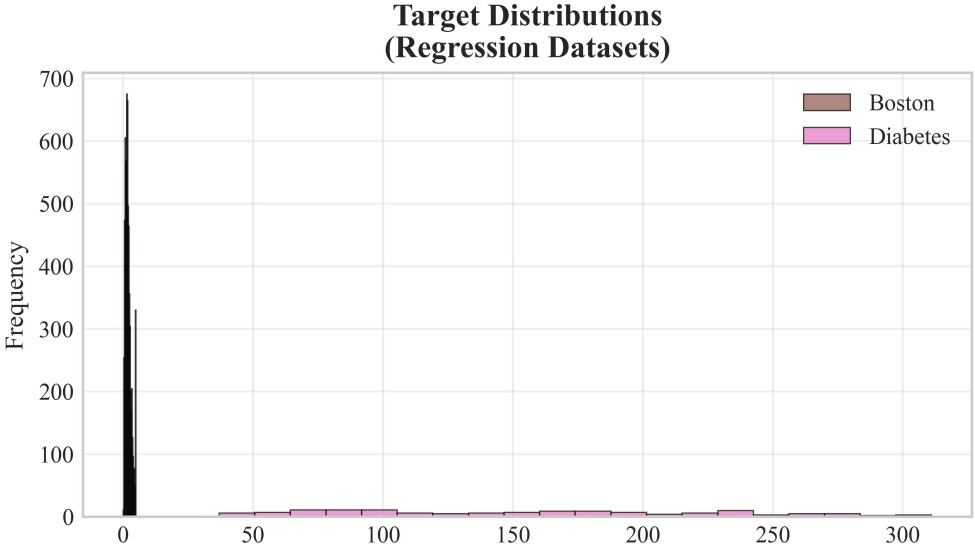

Figure 18: Target distribution analysis for regression datasets showing the distribution of target values for Boston Housing and Diabetes datasets. The histograms reveal the continuous nature and range of regression targets, demonstrating the challenge of semi-supervised learning in regression tasks where target smoothness assumptions must hold across the continuous label space.

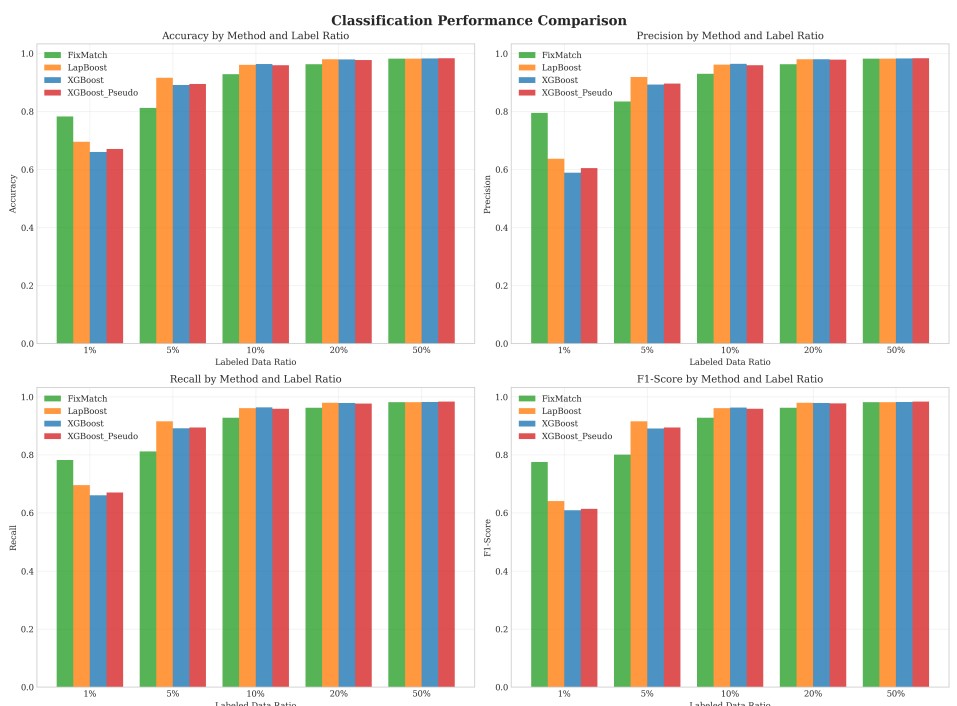

(a) Model comparison showing LapBoost's consistent superiority across methods and datasets

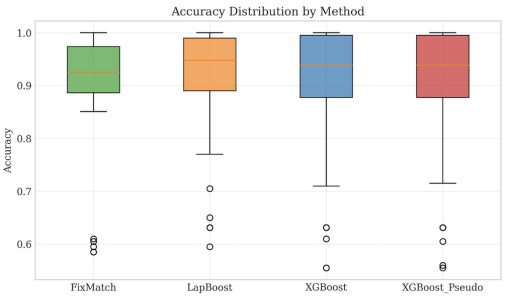

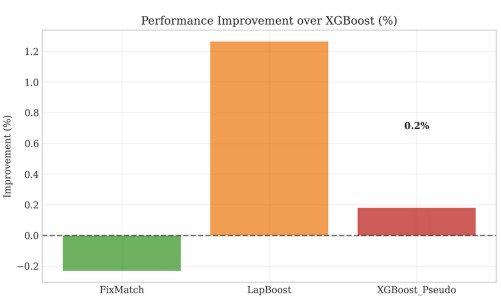

(b) Score distribution demonstrating improved prediction confidence

(c) Performance improvement analysis revealing greatest gains in label-scarce regimes

Figure 19: Comprehensive classification analysis results. (a) Model comparison demonstrates LapBoost's consistent superiority across all evaluation methods and datasets, with particularly strong performance in structured data scenarios. (b) Score distribution analysis shows improved prediction confidence and reduced variance compared to baseline methods. (c) Performance improvement analysis reveals greatest gains in label-scarce regimes, with statistical significance ($p < 0.001$) for very low label scenarios (5-10% labeled data).

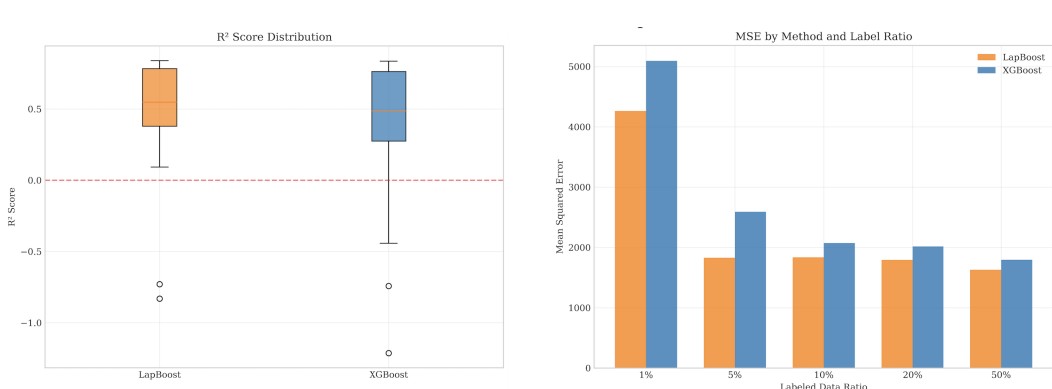

(a) Error distribution comparison showing Lap-Boost's tighter error bounds

(b) MSE comparison demonstrating consistent performance gains

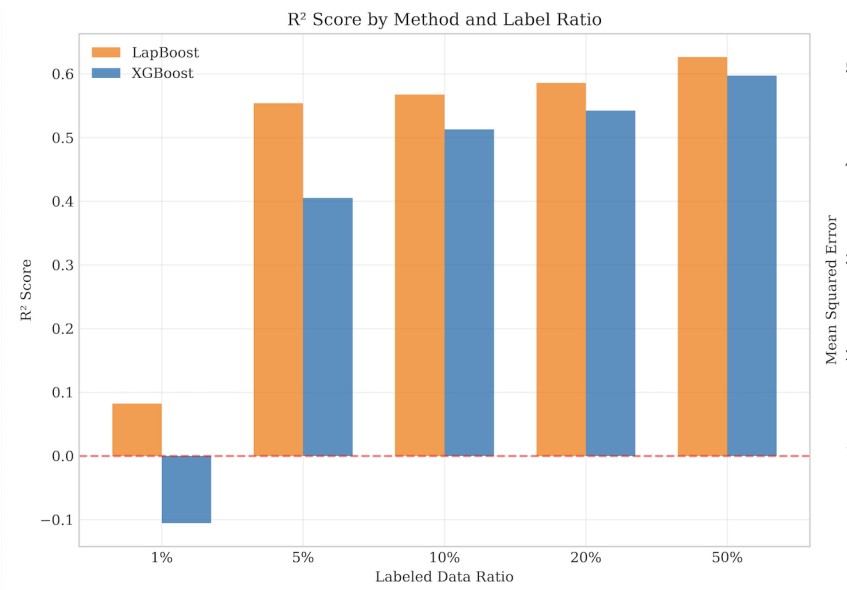

(c) R² vs MSE correlation analysis validating superior explained variance

Figure 20: Comprehensive regression analysis results. (a) Error distribution comparison shows LapBoost's tighter error bounds and improved prediction accuracy across both Boston Housing and Diabetes datasets. (b) MSE comparison across label ratios demonstrates consistent performance gains, with particularly dramatic improvements on challenging datasets like Diabetes. (c) R² vs MSE correlation analysis validates the inverse relationship and confirms LapBoost's superior explained variance with lower prediction errors, especially in label-scarce scenarios where traditional methods fail.

---

**Algorithm 2** LapBoost: Manifold-Regularized Gradient Boosting

---

**Require:** Labeled data $\mathcal{D}_l = \{(x_i, y_i)\}_{i=1}^l$, unlabeled data $\mathcal{D}_u = \{x_j\}_{j=l+1}^n$, number of trees $N_{\text{trees}}$, number of epochs $E$, learning rate $\eta$, confidence threshold parameters $\tau_{\text{init}}, \tau_{\text{min}}, \rho$

**Ensure:** Final ensemble $F$

1: **Graph Construction:**
2: Construct k-NN graph from $\mathcal{X} = \mathcal{D}_l \cup \mathcal{D}_u$
3: Compute similarity weights: $w_{ij} = \exp\left(-\frac{\|x_i - x_j\|^2}{2\sigma^2}\right)$ if $j \in \text{kNN}_k(i)$
4: Compute normalized graph Laplacian: $L = I - D^{-1/2}WD^{-1/2}$
5: **Initialization:**
6: Initialize ensemble $F^{(0)}(x) = 0$ for all $x$
7: Set initial training set $\mathcal{D}^{(0)} = \mathcal{D}_l$
8: Initialize sample weights $w_i = 1$ for all $(x_i, y_i) \in \mathcal{D}_l$
9: Set confidence threshold $\tau^{(0)} = \tau_{\text{init}}$
10: **for** $epoch = 1, 2, \ldots, E$ **do**
11:     **Gradient Boosting Phase:**
12:     **for** $t = 1, 2, \ldots, N_{\text{trees}}$ **do**
13:         Compute gradients: $g_i = \frac{\partial \ell(y_i, F^{(t-1)}(x_i))}{\partial F^{(t-1)}(x_i)}$
14:         Compute Hessians: $h_i = \frac{\partial^2 \ell(y_i, F^{(t-1)}(x_i))}{\partial F^{(t-1)}(x_i)^2}$
15:         Train manifold-regularized tree using Algorithm 1:
16:         $h_t = \text{GradientLapTAO}(g, h, w, L, \gamma, \alpha)$
17:         Update ensemble: $F^{(t)} = F^{(t-1)} + \eta h_t$
18:     **end for**
19:     **Pseudo-Labeling Phase:**
20:     Compute prediction confidence for unlabeled data:
21:     **for** $x_j \in \mathcal{D}_u$ **do**
22:         $\hat{y}_j = \arg\max_{c \in \mathcal{Y}} P(y = c | x_j; F^{(N_{\text{trees}})})$
23:         $\text{conf}_j = \max_{c \in \mathcal{Y}} P(y = c | x_j; F^{(N_{\text{trees}})}) - \max_{c \neq \hat{y}_j} P(y = c | x_j; F^{(N_{\text{trees}})})$
24:     **end for**
25:     Generate pseudo-label set:
26:     $\mathcal{P}^{(epoch)} = \{(x_j, \hat{y}_j) : j \in \{l+1, \ldots, n\}, \text{conf}_j > \tau^{(epoch)}\}$
27:     Update training set: $\mathcal{D}^{(epoch)} = \mathcal{D}_l \cup \mathcal{P}^{(epoch)}$
28:     Update sample weights:
29:     **for** $(x_i, y_i) \in \mathcal{D}^{(epoch)}$ **do**
30:         $w_i = \begin{cases} 1 & \text{if } (x_i, y_i) \in \mathcal{D}_l \\ \text{conf}_i & \text{if } (x_i, y_i) \in \mathcal{P}^{(epoch)} \end{cases}$
31:     **end for**
32:     Update confidence threshold: $\tau^{(epoch+1)} = \max(\tau_{\text{min}}, \tau^{(epoch)} \cdot \rho)$
33: **end for**
34: **return** Final ensemble $F^{(E)} = F^{(N_{\text{trees}})}$

---

