# OpenReview forum: "LapBoost: Graph Laplacian Regularized Gradient Boosting for Semi-Supervised Learning"
_ICLR.cc/2026/Conference — Submitted to ICLR 2026_

### Official Review · Reviewer_MdD3 · 2025-10-30

**Soundness:** 2
**Presentation:** 3
**Contribution:** 3
**Rating:** 4
**Confidence:** 3

**Summary:**

The paper proposes LapBoost, which integrates graph Laplacian regularization into gradient boosting by using LapTAO-style manifold-aware trees as base learners and a progressive, confidence-weighted pseudo-labeling schedule. Across multiple datasets and label budgets, the method reports consistent gains over supervised XGBoost and some SSL baselines, especially in low-label regimes.

**Strengths:**

1. Principled coupling of graph Laplacian regularization with gradient boosting via smoothed residual targets and manifold-aware trees.
2. Clear, modular algorithm that preserves GBDT residual-fitting behavior.
3. Consistent improvements in low-label regimes across several datasets; regression results are a nice addition rarely covered by SSL papers.
4. Useful practical guidance about complementarity between graph-based and consistency-based SSL paradigms.

**Weaknesses:**

1. Missing theory for the full procedure. No generalization or stability analysis for the ensemble with iterative pseudo-labeling and graph-smoothed gradients; even a monotone training-loss decrease proof or a bias–variance view would help.
2. Results may hinge on $k$, $\sigma$, and the distance metric; provide systematic ablations, automatic selection (e.g., local scaling), or robustness analyses under noisy neighborhoods/outliers.
3. The repeated solves in $\hat{r} = (H + \gamma L)^{-1}(-g)$ could be a bottleneck; show solver choices (CG/PCG, preconditioners), sparsity patterns, and scaling to larger $n$ (anchor graphs, ANN $k$-NN, graph sparsification).
4. Include LapMDBoost and label-propagation/label-spreading with a strong supervised learner, and consider recent tabular specialists (e.g., TabPFN) and shallow graph baselines (GCN over $k$-NN features) to contextualize gains.
5. Multiple $p$-values are reported across many conditions; discuss multiple-comparison control, effect sizes, and confidence intervals per dataset/label budget, not only aggregate tables.
6. The adaptive thresholding and weighting are intuitive, but more analysis of error propagation, calibration, and top-$k$ ensembling would increase trust; compare against simple self-training plugged into XGBoost under matched schedules.

**Questions:**

Please see Weaknesses.

---

### Official Review · Reviewer_PE5E · 2025-10-31

**Soundness:** 2
**Presentation:** 2
**Contribution:** 2
**Rating:** 2
**Confidence:** 4

**Summary:**

This manuscript proposes a semi-supervised learning approach that combines graph-based regularization with tree ensemble models, aiming to leverage the manifold structure of unlabeled data while benefiting from the strong discriminative power of gradient boosting. The idea of integrating these two paradigms is intuitively appealing and is supported by preliminary experiments that verify the feasibility of the proposed method. However, the work addresses a research direction that has seen limited recent interest in the broader machine learning community, and the experimental evaluation is relatively weak. Moreover, the method largely builds on existing components without significant algorithmic novelty or theoretical insight. Overall, the contribution appears incremental and does not advance the state of the art in a meaningful way.

**Strengths:**

1. The proposed LapBoost method effectively integrates graph Laplacian regularization into gradient boosting, demonstrating some performance gains in label-scarce settings.  It leverages the manifold structure of unlabeled samples while retaining the strong discriminative power of tree ensembles.

2. The approach is empirically validated and shows consistent improvements over supervised baselines under limited labeling.

3. The paper is well-structured and presents its ideas in a logically coherent manner.

**Weaknesses:**

1. The proposed method appears to be a relatively straightforward integration of existing techniques — graph Laplacian regularization and gradient boosting — without substantial algorithmic innovation or clear practical motivation. The problem setting itself is somewhat niche and lacks broad applicability in contemporary semi-supervised learning scenarios. Moreover, the work does not provide any theoretical analysis or guarantees (e.g., convergence, generalization bounds) to deepen understanding of the proposed integration.

2. The related work and references are somewhat outdated, with limited discussion of recent advances in semi-supervised tabular learning.

3. Several technical details are inadequately explained. For example, Figure 1 lacks sufficient annotation — its axes, data points, and key components are not properly described, making it difficult to interpret.

4. The experimental evaluation is limited in scope. The baselines are mostly classical supervised or basic semi-supervised methods, omitting comparisons to recent strong competitors. While LapBoost shows modest gains (often around 1%), it does not achieve consistent state-of-the-art performance across datasets or tasks.

5. For regression tasks, the work reports $R^2$ as the evaluation metric. Given that the values of $R^2$ are frequently around 0.5–0.6, it is unclear whether this performance level reflects a fundamental limitation of the problem or suboptimal model performance. Additional metrics (e.g., MAE, RMSE) would provide a more comprehensive assessment.

6. Key hyperparameter settings (e.g., graph construction parameters, regularization strength, boosting rounds) are either omitted or insufficiently detailed in the manuscript, hindering reproducibility of the work.

7. Finally, the presentation of results could be improved. A concise summary table that compares LapBoost against baselines across all key datasets and label rates would more clearly demonstrate the method’s effectiveness (or limitations).

**Questions:**

How to get the “Robustness Score” reported in Table 2？

---

### Official Review · Reviewer_1y87 · 2025-11-01

**Soundness:** 2
**Presentation:** 3
**Contribution:** 2
**Rating:** 2
**Confidence:** 4

**Summary:**

This paper presents a boosting method for semi-supervised learning that integrates graph-based regularization with gradient boosting that uses Laplacian-regularized Tree-based Alternating Optimization as base learners. Comprehensive experiments on five classification tasks and two regression tasks are conducted to show the improved performance over other three approaches  as baselines.

**Strengths:**

Good combination of graph-based regularization with modern gradient boosting

**Weaknesses:**

This work doesn't reflect the paradigm shift in the era of large language/speech/vison/muti-modal models. The experiments should include the results that use domain unlabeled data and labeled data to fine tune a serious of base large models. Whether the proposed approach is still competitive.

**Questions:**

N/A

---

### Official Review · Reviewer_abmd · 2025-11-02

**Soundness:** 2
**Presentation:** 1
**Contribution:** 2
**Rating:** 2
**Confidence:** 3

**Summary:**

This paper proposes a semi-supervised ensemble method LapBoost that combines the sequential residual fitting in gradient-boosting and the manifold regularization in graph Laplacian methods for semi-supervised learning (SSL). It borrows from the alternating optimization procedure in LapTAO, and fits decision trees on sequential pseudo-residuals that are smooth on the graph.  After the training of gradient-boosted trees,  a pseudo-labeling step is implemented to increase the labeled set by adding unlabelled samples with confidently estimated labels. Then the training of LapBoost restarts on the newly augmented labeled set. This process repeats itself after a number of epochs with decreasing threshold on the confidence level for pseudo-labeling, before outputting the predictive model obtained on the final labeled set. The proposed method achieves a higher overall performance on a range of datasets over its supervised counterpart XGBoost. Two semi-supervised methods, FixMatch and XGBoost with pseudo-labeling, are also tested.

**Strengths:**

- The idea of combining XGBoost and graph Laplacian regularization to improve the learning performance on partially labelled tabular data is interesting and well motivated.

- The proposed method is tested on several real datasets.

**Weaknesses:**

- The presentation quality is poor. It is not easy to reconstruct the proposed algorithm from its description in Section 3, where different elements of the algorithm are presented without explaining clearly how they are combined together. The workflow presented in Figure 1b seems to contradict the textual description of the algorithm: in the workflow the XGBoost training appears to precede the graph regularization, while according to the text the graph regularization is incorporated into the XGBoost training. Moreover, the workflow does not convey that the pseudo-labeling step belongs to a loop outside the loop of the XGBoost training, meaning that it follows the end of a XGBoost training process and leads to a new cycle. The notations that appear before their definitions add to the confusion. For instance, the sample weights $\omega_i$ appear in (2) before being defined in (4), and the fact that the same symbol $\omega$ is used for the sample weights $\omega_i$ in the empirical loss and the connection weights $\omega_{i,j}$ on the graph does not help either. The notation $\mathcal{D}^{(t)}$ standing for the training set at the $t$-th epoch does not seem to be defined in the manuscript, I had to go to the pseudo-code in the appendix to parse its meaning.

- The experimental results are reported in an unconventional manner. Even though many real datasets were tested, the authors chose to focus on the overall performance, rather than discussing the specific performance on each dataset. This could hide valuable information on how different semi-supervised methods compare to each other on various tasks, while the authors claimed to "offer practical guidance for method selection based on data characteristics". Even in Table 5 in the appendix where the data-specific performance is reported, there are only the results on three data sets of the five data sets that were tested, and no result for XGBoost with pseudo-labeling. The experimental results are not very convincing either. Even with respect to the overall performance, the proposed method is hardly competitive compared to the other two semi-supervised methods. The range of compared semi-supervised baselines is also limited, including none of the numerous graph-based SSL methods in the literature.

- The manuscript contains some overstatements, such as the claim that this article "provides the first systematic characterization of when different SSL approaches should be applied" in the abstract, while no theoretical analysis is conducted in this paper and there have already been many empirical studies that compared different SSL methods.

**Questions:**

- What is the relation between the manifold-regularized objective $\mathcal{L}^{(t)}$ in (2) and the smoothness term $\mathcal{L}_{\rm grad-smooth}$ in (3)? And how to understand them intuitively?

- Could you match the workflow in Figure 1b to the pseudo-code in Algorithms 1&2?

- The outside loop created by the pseudo-labeling step increases greatly the computational time. To what extent the pseudo-labeling is critical to the performance of the proposed method?

- What did you mean exactly when you said that the finding of this paper offered "the first systematic characterization of when different SSL approaches should be applied"?

---

### Meta-Review · Area_Chair_XMWF · 2025-12-07

**Summary:**

After careful review of the revised manuscript and the authors' point-by-point responses, I have decided to recommend rejection.

While I appreciate the authors’ thorough efforts in addressing the detailed feedback from all four reviewers, key issues remain unresolved. Specifically, the explanations regarding the paper’s limited novelty, experimental evaluation design, lack of convergence analysis, and sensitivity to hyperparameters were not convincing. The authors claimed these points had been supplemented in the updated version, but I did not find substantial additions on these matters in the document.

I would have been open to accepting the paper even with its initially low scores if the revised manuscript had aligned closely with the commitments made in the rebuttal. However, after carefully reviewing the updated submission, I find that the changes made still fall significantly short of what was described and promised in the response letter.

Given that these concerns go to the core of the paper’s technical soundness and contribution, I do not believe it meets the acceptance criteria in its current form.

**Reviewer Concerns:**

Most minor issues have been addressed, while key issues regarding **limited novelty**, **experimental evaluation design**, **lack of convergence analysis**, and **sensitivity to hyperparameters** remain unresolved.

**Reviewer Scores:**

Scores: 2, 4, 2, 2

Most reviewers may maintain their orignal scores.

---

### Decision · Program_Chairs · 2026-01-26

Reject